# Gut Microbiota-Derived Metabolites and Cardiovascular Disease Risk: A Systematic Review of Prospective Cohort Studies

**DOI:** 10.3390/nu14132654

**Published:** 2022-06-27

**Authors:** Raul Sanchez-Gimenez, Wahiba Ahmed-Khodja, Yesica Molina, Oscar M. Peiró, Gil Bonet, Anna Carrasquer, George A. Fragkiadakis, Mònica Bulló, Alfredo Bardaji, Christopher Papandreou

**Affiliations:** 1Department of Cardiology, Joan XXIII University Hospital, 43005 Tarragona, Spain; raul.sagi@hotmail.com (R.S.-G.); opi220290@gmail.com (O.M.P.); gil.bonet.p@gmail.com (G.B.); carrasquer1987@gmail.com (A.C.); abardaji.hj23.ics@gencat.cat (A.B.); 2Pere Virgili Health Research Institute (IISPV), 43005 Reus, Spain; wahiba.ahmed-khodja@urv.cat (W.A.-K.); ymolinac8@gmail.com (Y.M.); 3Department of Medicine and Surgery, Rovira i Virgili University, 43003 Tarragona, Spain; 4Department of Biochemistry and Biotechnology, Rovira i Virgili University, 43201 Reus, Spain; 5Department of Nutrition and Dietetics Sciences, School of Health Sciences, Hellenic Mediterranean University, 72300 Siteia, Greece; fragkiadakis@hmu.gr; 6CIBER Physiology of Obesity and Nutrition (CIBEROBN), Carlos III Health Institute, 28029 Madrid, Spain

**Keywords:** gut microbiota, metabolites, cardiovascular disease, mortality

## Abstract

Gut microbiota-derived metabolites have recently attracted considerable attention due to their role in host-microbial crosstalk and their link with cardiovascular health. The MEDLINE-PubMed and Elsevier’s Scopus databases were searched up to June 2022 for studies evaluating the association of baseline circulating levels of trimethylamine N-oxide (TMAO), secondary bile acids, short-chain fatty acids (SCFAs), branched-chain amino acids (BCAAs), tryptophan and indole derivatives, with risk of cardiovascular disease (CVD). A total of twenty-one studies were included in the systematic review after evaluating 1210 non-duplicate records. There were nineteen of the twenty-one studies that were cohort studies and two studies had a nested case–control design. All of the included studies were of high quality according to the “Newcastle–Ottawa Scale”. TMAO was positively associated with adverse cardiovascular events and CVD/all-cause mortality in some, but not all of the included studies. Bile acids were associated with atrial fibrillation and CVD/all-cause mortality, but not with CVD. Positive associations were found between BCAAs and CVD, and between indole derivatives and major adverse cardiovascular events, while a negative association was reported between tryptophan and all-cause mortality. No studies examining the relationship between SCFAs and CVD risk were identified. Evidence from prospective studies included in the systematic review supports a role of microbial metabolites in CVD.

## 1. Introduction

Cardiovascular disease (CVD) remains a major public health issue [1]. The progress in its prevention and management depends on a better understanding of the mechanisms underlying disease development. Identification of circulating biomarkers with prognostic value may help to both identify pathophysiological processes relevant to CVD development and improve preventive cardiovascular risk reduction efforts [2]. Recent development of omics technologies has improved biomarker discovery, leading to the identification of a number of disease-associated circulating molecules [3]. Metabolomics has considerably increased interest in metabolism across cardiovascular research [4] and identifying metabolites associated with disease risk can further highlight the critical metabolic pathways in CVD etiology.

Gut microbiota-derived metabolites like trimethylamine N-oxide (TMAO), secondary bile acids, short-chain fatty acids (SCFAs), branched-chain amino acids (BCAAs), tryptophan and indole derivatives have recently attracted considerable attention due to their role in host-microbial crosstalk [5,6] and their link with health. Numerous observational studies have found associations between circulating microbial metabolites and CVD risk [7]. Since the discovery of a relationship between the gut microbiota-dependent metabolite TMAO and CVD, in 2011 [8], nine systematic reviews of prospective studies published up to 2020, suggest a strong association between elevated circulating levels of TMAO with risk of several CV outcomes [9,10,11,12,13,14,15,16,17]. Since then, the number of prospective TMAO profiling studies on CVD risk has substantially increased [18,19,20,21,22,23,24,25,26,27]. However, less evidence exists on other microbial-related metabolites in relation to incident CVD. In this sense, two previous systematic reviews of prospective studies on BCAAs, found higher levels associated with increased risk of CVD [9] and ischemic stroke [16], whereas tryptophan levels were inversely associated with CVD risk [9]. Regarding secondary bile acids, SCFAs and indole derivatives, evidence from some, but not all, prospective studies suggest that these metabolites are associated with CV outcomes [9,28,29,30,31,32], and to our knowledge, no systematic review has been conducted on these metabolites yet.

Gut microbiota-derived metabolites are produced from bacterial metabolism of dietary substrates, modification of host molecules, such as bile acids, or directly from bacteria [33]. Summarizing the available evidence on the role of these metabolites on CVD risk could assist in understanding better the metabolic perturbations associated with CVD development. Therefore, we conducted a systematic review for the association of circulating levels of microbial metabolites with CVD incidence.

## 2. Materials and Methods

This systematic review was conducted following the Preferred Reporting Items for Systematic reviews and Meta-Analyses (PRISMA) guidelines (http://www.prisma-statement.org/ (accessed on 2 February 2022) [34,35] and the review protocol was registered with PROSPERO (PROSPERO 2022: CRD42021291322).

### 2.1. Eligibility Criteria

We considered: (1) prospective observational studies (including cohort, case-cohort or nested case-control) conducted in participants at least 18 years of age; (2) studies published in English and not included in previous systematic reviews performed in this topic before [9,10,12,13,15,16,17,36]; (3) studies including TMAO, secondary bile acids, SCFAs, BCAAs, tryptophan and indole derivatives as main exposures; (4) metabolites measured in serum or plasma (we excluded metabolomics profiling conducted in urine samples); (5) metabolite profiling conducted at baseline in the context of a prospective study; (6) studies that included the following end-points: coronary artery disease (CAD)/coronary heart disease, myocardial infarction (MI), stroke, atrial fibrillation (AF), heart failure (HF), peripheral artery disease (PAD), major adverse cardiovascular event (MACE) and/or death (either cardiovascular or all-cause mortality); (7) studies that reported a measure of association (e.g., odds ratio [OR], hazard ratio [HR], risk ratio [RR] with their corresponding 95% confidence intervals [CI]).

Studies were excluded if they were cross-sectional, interventional, case reports, comments, letters, editorials, duplicate studies, narrative or systematic reviews and study protocols. When there was more than one study from the same cohort, the publication with the longer follow-up or more incident cases was selected.

### 2.2. Literature Search

A literature search was carried out using the MEDLINE-PubMed and Elsevier’s Scopus databases from inception through 9 June 2022. For the database searches, terms related to TMAO, bile acids, SCFAs, BCAAs, tryptophan and indole derivatives were combined with terms related to CVDs (Appendix A).

### 2.3. Screening and Data Extraction

Screening of eligible studies and data extraction were performed independently by three independent reviewers (R.S.G. W.A-K., and Y.M.), and disagreement between the three authors was resolved by consulting senior researchers (C.P. and A.B.).

The data extracted from each eligible study included the following: first author’s surname, year of publication, country where the study was conducted, title, journal, design of the study, sample size, duration of follow-up, characteristics of participants, assay method, number, type, and identity of metabolites investigated, biospecimen (serum/plasma), fasting status, statistical approach, covariates included in the fully adjusted model and major findings.

### 2.4. Quality Assessment

The same investigators (R.S.G., W.A-K., and Y.M.) independently assessed each study for methodological quality using the Newcastle–Ottawa Scale (NOS) for cohort and nested case-control studies [37]. This assessment tool consists of eight items, and divides the studies with a scale of scores of 0 to 9 from poor to high quality, respectively. The maximum score is 9, and a score of ≥ 6 indicates high methodological quality.

## 3. Results

### 3.1. Selection of Studies

#### Literature Search

The initial search yielded 1210 non-duplicate records which were screened and 1137 records were removed based on title and abstract, leaving 73 reports for full-text examination (Figure 1). We excluded 52 reports for reasons indicated in Figure 1 and finally 21 studies were deemed eligible for inclusion in the present systematic review. Of them, 19 included prospective cohorts [18,19,20,22,24,25,26,27,28,29,30,31,36,38,39,40,41], one study included one nested case-control study [23] and one study included two nested case–control studies [21]. All of the included studies were of high quality with a mean of 8.9 points in the Newcastle–Ottawa scale (range: 7–9) (Appendix A).

### 3.2. Study Characteristics

The main characteristics of the 21 selected articles [18,19,20,21,22,23,24,25,26,27,28,29,30,31,32,36,38,39,40,41,42] are summarized in Table 1. Most of them (n = 18) were published in the last five years (2022–2018), while none were published before 2015. A total of ten of the studies were conducted in the USA, seven in Europe, two in Taiwan and two in China. All studies used a cohort design except two that included a nested case-control design [21,23]. There was one article that included three cohorts [19] whereas two articles included two cohorts as the discovery and replication sample analyses [28,40]. The minimum and maximum follow-up was 1 month and 28 years, respectively, while in most of the studies the average follow-up was up to 10 years. In total, the studies included 58,691 participants and the number of cases in the single studies ranged from 32 to 4791. In 14 articles participants were free of CVD at baseline [18,21,22,23,24,27,28,29,30,31,32,36,39,40] but two of them were conducted in patients with type 2 diabetes (T2D) [18,22] and two studies in chronic kidney disease (CKD) patients [31,32]. On the other hand, in seven articles participants had at least one CVD condition [19,20,25,26,38,41,42]. The main outcome in most of the studies was all-cause and CVD mortality, followed by incident MACE, AF, HF, stroke, and additional CV outcomes. A total of six articles analyzed metabolites in serum [25,29,30,31,32,36] and the rest in plasma. In 13 articles blood sample was collected in fasting [21,24,26,27,28,29,30,31,32,36,38,39,40] and in five articles in non-fasting conditions [18,22,25,41,42], while in two articles the fasting status was not reported [19,20] and in one article participants provided blood samples in either fasting or non-fasting conditions [23]. In 20 studies, the analyses of metabolites were performed by using mass spectrometry (MS) coupled to different chromatography techniques, while in one study a nuclear magnetic resonance (NMR) platform was used [40].

### 3.3. Metabolites Associated with CVD Risk

Among the 21 selected studies, 15 [18,21,22,23,24,25,26,28,29,31,32,38,39,41,42] examined associations of microbial metabolites with risk of CVD, CAD, MACE, major vascular event, AF, HF, MI, stroke, coronary revascularization and a composite event of death or HF-related re-hospitalization.

#### 3.3.1. Total CVD

Lee et al. used time-varying cumulative averaging of serial TMAO measures (assessed at baseline and 7 years) in relation to risk of CVD in 4131 participants free of CVD [24] During a median follow-up of 15 years, 1766 CVD events occurred. After extensive adjustment for demographics, traditional CVD risk factors, medications, diet and renal function, no significant associations were observed.

Similarly, in the Winther et al. study of 311 patients with T2D but free of CVD no significant associations were found between TMAO and risk of CVD (116 cases after 6.5 years) after adjustment for demographics, traditional factors for CVD and renal function markers [18].

With regards to bile acids, Cheng and colleagues performed metabolite profiling (LC-MS) on plasma samples collected from 2327 participants of the Framingham Offspring Study who underwent a routine examination between 1991 and 1995 and were followed up to 20 years [28]. During this period, 358 individuals developed CVD. Of the 217 metabolites analyzed, three secondary bile acids (glycocholate, glycodeoxycholates, deoxycholates) were not significantly associated with CVD [28].

Another secondary bile acid, the deoxycholic acid was also not significantly associated with risk of CVD in 3147 patients with chronic kidney disease after the inclusion of many potential confounders in the analyses conducted by Frazier et al. [32].

On the other hand, in a large cohort of women free of CVD (Women’s Health Study) 27,041 subjects who were followed over a mean 18.6 years elevated baseline plasma levels of total BCAAs assessed by NMR were found associated with higher risk of CVD (per 1SD increase; HR 1.14 [95% CI 1.08–1.18]) [39].

#### 3.3.2. MACE

The MACE components varied by studies and a detailed description can been seen in Table 1.

In 1463 patients with T2D, associations between TMAO plasma concentrations and MACE were investigated by Croyal et al. [22]. During a follow-up of 7.1 years, 403 MACE cases were ascertained. In multivariate Cox regression models (covariates: demographics, personal history of MI, renal function markers, NT-proBNP), compared with the first quartile, those patients in the fourth quartile of TMAO concentrations were at increased risk for MACE (HR 1.29 [95% CI 1.02–1.64]) [22].

In another prospective analysis of plasma TMAO conducted in 262 symptomatic PAD patients by Roncal et al., during a mean period of 4 years, 135 cases were recorded. Multivariable Cox regression analyses (covariates: demographics, CRP, smoking, T2D, hypertension, dyslipidemia, HDL-C, renal function) did not reveal significant associations between one log unit increase in TMAO concentrations and risk of MACE [38].

In a study by Fan et al., 147 patients with CKD stage 1–5 were followed over a 3-year period, and 47 of them developed MACE [31]. Elevated levels of serum indoxyl sulfate were associated with an increased risk of MACE even after adjustment for demographics, traditional CVD risk factors, medications, renal function markers (HR 1.45 [95% CI 1.02–2.06]) [31].

#### 3.3.3. CAD, Major Vascular Event, AF, HF, MI, Stroke, Coronary Revascularization

Liu et al. analyzed data from two cohorts, the Nurses’ Health Study II and the Health Professionals Follow-up Study, 1216 participants free of CVD and T2D were followed and 608 developed CAD. TMAO was not independently associated with CAD risk in pooled analysis [23].

In another recent study by Papandreou, C et al., associations of plasma TMAO and its precursors with incident AF and HF were examined among 1879 subjects at high CV risk [21]. A total of two nested case-control studies were conducted within the PREDIMED study and after a mean follow-up for about 10 years, 512 AF and 334 HF incident cases were ascertained. Contrary to choline, betaine and dimethylglycine, TMAO was not associated with AF and HF [21].

TMAO was also not significantly associated with MI in a cohort of 1726 patients with suspected functionally relevant CAD [25].

Contrary to these findings, a more recent study by Chen et al. revealed that TMAO levels above the median were associated with higher risk of a major vascular event defined as a composite of transient ischemic attack, recurrent ischemic stroke, and MI in 291 patients with ischemic stroke [26].

In the Atherosclerosis Risk in Communities Study (ARIC) 3922 participants were included in the analyses performed by Alonso et al. and followed for over 20 years, 608 AF incident cases were reported [31]. MS measurements of 245 metabolites in serum were performed. Among two secondary bile acids measured (glycocholenate sulfate, glycolithocolate sulfate) only glycocholenate sulfate was associated with AF (per SD, HR 1.12 [95% CI 1.04–1.21]) [29]. Deoxycholic acid, however, was not associated with HF events in the CRIC study [32].

In another study (Du et al.) of 138 patients with ST-segment elevation MI with acute HF, 16 plasma amino acids were measured with MS. During a follow-up of 3 years, 32 deaths and 21 hospitalizations for HF were registered [41]. Multivariable Cox regression analysis revealed that increased plasma concentrations of BCAAs were associated with a composite of death and HF hospitalization (per SD HR 2.67 [95% CI: 2.41–4.41]) [41].

In the Women’s Health Study, Tobias et al. found increased total BCAAs plasma concentrations associated with MI (HR 1.16 [95% CI 1.06–1.26]) and coronary revascularization (HR 1.17 [95% CI, 1.11–1.25]) [39].

In the Wang et al. study, 136 patients with HF and 51 participants without HF were followed for a mean of 2.3 years and 35 of them had a composite event of death or HF-related re-hospitalization [42]. In the multivariable Cox regression analysis, plasma concentrations of indoxyl sulfate measured by MS were not significantly associated with the composite event.

### 3.4. Metabolites Associated with CVD Mortality and All-Cause Mortality

Eleven of the included studies in this systematic review (Table 1) examined the relationship of microbial metabolites and risk of CVD mortality and/or all-cause mortality [18,19,20,22,25,27,28,30,32,36,40]. Six, four and one of the studies examined the relationship of TMAO [18,19,20,22,25,27,38], bile acids [28,30,32,39] and tryptophan [41], respectively, with these outcomes.

#### 3.4.1. TMAO

In a very recent study by Amrein et al., higher TMAO concentrations were associated with all-cause mortality (per log2-transformed increase; HR 1.19 [95% CI 1.01–1.40]) but not CVD-mortality (HR 1.11 [95% CI 0.99–1.26]) [25].

Significant associations were also reported lately in a large cohort of 5331 participants. During 13.2 years, 4791 deaths were recorded and after adjusting for many potential confounders increased plasma levels of TMAO were associated with an increased risk of all-cause mortality (HR 1.30 [95% CI 1.17–1.44]) [27].

In the Winther et al. study after a median follow-up of 6.8 years and 6.5 years, 106 total deaths and 44 CVD deaths were reported, respectively [18]. No significant associations between one SD increase in TMAO plasma concentrations and any of these outcomes were found in the fully adjusted model. Likewise, in another cohort study conducted by Croyal et al. 1463 patients with T2D were followed for 7.1 years and 538 death cases were recorded. Plasma TMAO was not independently associated with risk of all-cause mortality [22].

In contrast, Ringel et al. investigated associations of plasma TMAO with short- (30 days) and long-term (9 years) all-cause mortality in three independent cohorts of 1666 patients with stable atherosclerotic CVD (449 patients with CVD in a cohort of 904 patients), acute MI (326 patients), cardiogenic shock (436 patients), and T2D (28.7%, 27.3%, 32.4% respectively) [19]. After 1 month, 194 deaths were reported in the CULPRIT-SHOCK cohort, whereas after 9 years, 99 deaths were reported in the LIFE-CAD cohort and 26 deaths in the AMI cohort. Significant associations between TMAO and mortality were found in the LIFE-CAD study of patients with suspected coronary artery disease (multivariate-adjusted HR 1.24 [95% CI 1.01–1.51]) [19]. Similarly, in another cohort study conducted by Israr et al. in 806 acute HF patients, plasma TMAO was quantified and associated with short- (30 days) and long-term (1 year) all-cause mortality [20]. Multivariate analysis revealed significant associations between increased plasma concentrations of TMAO with risk of mortality after 30 days (62 deaths; HR 1.39 [95% CI 1.05–1.84]) and 1 year (213 deaths; HR 1.26 [95% CI 1.08–1.47]) [20].

Finally, Roncal et al. determined TMAO in plasma using MS in 262 patients with symptomatic PAD who were followed for a mean period of 4 years [38]. During this period, 101 all-cause and 39 CVD mortality cases were reported. A significant association between high TMAO levels and all-cause mortality was found in the unadjusted model, but not in the multivariate-adjusted model. However, higher TMAO levels were associated with CVD-mortality (per one SD increase; HR 1.52 [95% CI 1.27–1.82]) and also when TMAO was treated as a binary variable (> 2.26 µmol/L vs. <2.26 µmol/L; HR 3.36 [95% CI 1.68–6.70]) [38].

#### 3.4.2. Secondary Bile Acids

Yu et al. analyzed 204 serum metabolites in 1887 African Americans from the ARIC Study using MS techniques [36]. During a median follow-up period of 22.5 years 671 deaths were reported. Glycocholate was the only bile acid independently associated with the risk of CVD mortality (HR 1.14 [95% CI 1.07–1.22]) and all-cause mortality (HR 1.12 [95% CI 1.07–1.16]) [36].

Significant results were also obtained for the relationship between deoxycholic acid and all-cause mortality (HR 2.13 [95% CI 1.25–3.64]) in patients with chronic kidney disease [32].

On the other hand, Huang et al. analyzed data from the Alpha-Tocopherol, Beta-Carotene Cancer Prevention (ATBC) Study Cohort of 620 men free of CVD, who were followed for 28 years. Of the 406 metabolites quantified in serum using MS techniques no significant associations of the ten secondary bile acids analyzed with CVD or all-cause mortality were observed [30].

No significant associations between three secondary bile acids and all-cause mortality were also reported in an earlier study by Cheng et al. among 2327 participants free of CVD [28].

#### 3.4.3. Tryptophan and Indole Derivatives

Balasubramanian et al. used two datasets created from the Women’s Health Initiative study consisting of 943 subjects (discovery set; 417 deaths) and 1355 subjects (replication set; 685 deaths) free of CVD [40]. Of the 470 metabolites measured in plasma with LC-MS/MS the amino acid, tryptophan was inversely associated with all-cause mortality in the discovery (HR 0.82 [95% CI 0.75–0.89]) and replication set (HR 0.87 [95% CI 0.81–0.94]) [40]. No studies examining associations between indole derivatives and mortality were identified in the present review, and more work is needed in this area.

## 4. Discussion

This systematic review summarized the evidence of the associations between circulating microbial-related metabolite levels and risk of CVD. Associations of TMAO and subsequent risk of CV outcomes were supported by some [19,20,22] but not all prospective studies [18,21,22,38]. Inconsistent results were also obtained for secondary bile acids in relation to CVD and related outcomes [30,31], and CVD/all-cause mortality [28,30,36]. With regards to BCAAs [39,41], their associations with CV outcomes were robust among the studies, whereas one study reported a negative relationship between tryptophan and mortality [40] and another study revealed an association between indole derivatives and MACE [30].

### 4.1. TMAO

In recent years, several systematic reviews of prospective studies have suggested that TMAO is an independent risk factor for CVD [9,10,11,12,13,14,15,16,17]. Several mechanisms have been described in the literature that could explain these associations. TMAO production from ingestion of animal foods such as red meat may affect cholesterol and lipid metabolism, and endothelial dysfunction as well as platelet activity, which could result in atherosclerosis development [8] the dominant cause of CVD [43]. On the other hand, a recent Mendelian randomization analysis examining the causal direction between TMAO and cardiometabolic diseases suggested that CVD may increase circulating TMAO levels and that observational evidence for CVD may be due to confounding or reverse causality [44].

Our updated systematic review included a total of 11 individual studies that examined the associations of TMAO with CV outcomes. Among them, four outcomes (MACE, major vascular event, CVD mortality, all-cause mortality) were found to be positively associated with TMAO levels in some but not all of the included studies. In three previous meta-analyses of 19, [10], 11 [12] and 17 [11] prospective cohort studies published in 2017, elevated circulating levels of TMAO were associated with higher risk of MACE and all-cause mortality with low to moderate and moderate to high heterogeneity, respectively. A more recent systematic review with meta-analysis of three cohort studies also found moderate heterogeneity across the studies examining associations between TMAO and MACE incidence with no evidence of heterogeneity in the association between TMAO and all-cause mortality [13]. Another recent meta-analysis of nine cohort studies revealed that elevated plasma TMAO concentrations were associated with higher risk of MACE evidence of moderate heterogeneity [17]. In our systematic review the discrepancies in the results of the included studies may be due to differences in: disease status, study geographical locations, number of cases, fasting status at blood draw, adjustments for confounders and follow-up length. Disease status might affect the association between TMAO and CV outcomes through affecting microbiota composition which regulate the circulating TMAO levels [45,46]. The role of geographical location in the above associations could be possibly due to the role of diet in regulation of blood TMAO levels [47,48]. Variation in the number of cases among the included studies may have affected statistical power to detect associations. In two studies, blood samples were collected in non-fasting conditions, while in two others fasting status was not indicated which might have affected TMAO concentrations. Finally, uncontrolled confounding factors such as dietary habits, gut microbiota composition and genetic variation may significantly affect the concentrations of TMAO representing an important source of variability in interpretations.

### 4.2. BCAAs

The findings of the present review confirm previous findings of a former systematic review on the strong positive prospective associations between BCAAs and CVD risk [9]. In Tobias et al. study [39], a significant positive association was observed between BCAAs with long-term CVD risk, comparable in magnitude to LDL cholesterol, an established CVD biomarker. Notably, this relationship was prominent among women who developed T2D prior to CVD. Also, in Du et al. [41], increased plasma BCAAs levels were associated with long-term adverse cardiac events in patients with STEMI and AHF and additionally, BCAAs improved the predictive value of NT-proBNP and the GRACE score. These consistent findings reinforce the role of these metabolites as potential biomarkers of biological dysfunction related to CVD.

Potential mechanisms explaining the relationship between elevated BCAAs levels and risk of CVD include activation of the mammalian target of rapamycin (mTOR) signaling, which leads to exacerbated cardiac dysfunction, and remodeling in myocardial infarction [49].

### 4.3. Secondary Bile Acids

To our knowledge, this is the first systematic review of secondary bile acids in relation to CVD risk. Inconsistent results were obtained for these relationships. No significant results were shown for total CVD [30,34], whereas increased levels of glycocholenate sulfate and glycocholate were associated with risk of AF [31], CVD mortality and all-cause mortality [11], respectively. However, the associations of glycocholate, and several other secondary bile acids with mortality were not confirmed in another three studies [27,30,32]. Potential mechanisms underlying the relationship between glycocholenate and AF may include potential arrhythmogenic effects of this bile acid. With regards to glycocholate, is an indirect cholesterol-derived bile acid involved in cholesterol metabolism and is it considered to be a key factor for mortality [50]? Another bile acid, the deoxycholic acid, was also associated with mortality [32]. Elevated levels of this bile acid may contribute to inflammation and immune dysregulation, and vascular calcification and may hasten chronic kidney disease progression.

### 4.4. Tryptophan and Indole Derivatives

In our systematic review tryptophan was inversely associated with all-cause mortality (Women’s Health Initiative study) [40]. Tryptophan is an essential amino acid vital for maintaining health and homeostasis [51]. Given that a higher kynurenine/tryptophan ratio reflects an increased indoleamine 2,3-dioxygenase activity and predicts mortality [52] we can speculate that the effect of tryptophan on mortality might depends on its degradation rate to kynurenine and thus higher circulating tryptophan levels may be due to a low rate of conversion.

Of indole derivatives, indoxyl sulfate is a protein-binding molecule that exhibits CV toxicity and in our systematic review was associated with risk of MACE in CKD patients [31]. Some prospective studies suggested that elevated circulating indoxyl sulfate levels may have a significant role in the vascular dysfunction in CKD patients [53]. When dietary tryptophan is metabolized into indole by intestinal bacteria and absorbed, then it is converted to indoxyl sulfate in the liver. In patients with impaired renal function indoxyl sulfate is not removed effectively [54] and accumulates in the blood exerting its deleterious effects on the endothelium by increasing vascular calcification and stiffness [55,56].

### 4.5. SCFA

Altered circulating SCFA profiles may be related to several metabolic conditions. SCFA are produced in the gut by the metabolic activity of the intestinal microbiota as catabolic end-products from the fermentation of undigested dietary components, mainly complex carbohydrates. Although studies have focused on different clinical conditions, evidence on the role of these mediators in CVD is lacking [57].

### 4.6. Strengths and Limitations

To the best of our knowledge, this systematic review provided the most comprehensive evidence on the associations of microbial metabolites with risk of several CV outcomes. Focusing on prospective studies allowed us to limit the influence of reverse causality and selection bias, and all of the included studies were evaluated to be of high quality, as assessed with the “Newcastle–Ottawa Scale”.

There are several limitations to be considered. First, the lack of studies on SCFAs limited conclusions about the role of main metabolites produced by intestinal bacteria in CVD and this is an area that needs development. Second, due to the observational nature of the studies included in the review, potential causal mechanisms underlying the above associations cannot be inferred. Third, confounder adjustment differed across the studies, which may have resulted in different associations between metabolites such as TMAO or bile acids and CV outcomes. Fourth, we were unable to conduct a meta-analysis, mainly because of the heterogeneity of the included articles. Differences in metabolite targets, characteristics of the study populations and differential reporting of associations could increase the possibility the results of the meta-analysis not to be useful and meaningful.

## 5. Conclusions

In conclusion, our systematic review summarized the evidence on the associations of gut microbiota-derived metabolites with CV outcomes and showed inconsistent results for TMAO and bile acids but robust ones for the relationships between BCAAs and CVD. There are still a small number of longitudinal studies assessing the association of tryptophan and indole derivatives with the risk of CV outcomes. Further studies are needed to investigate whether circulating microbial metabolites could be an intervention target for CVD.

## Figures and Tables

**Figure 1 nutrients-14-02654-f001:**
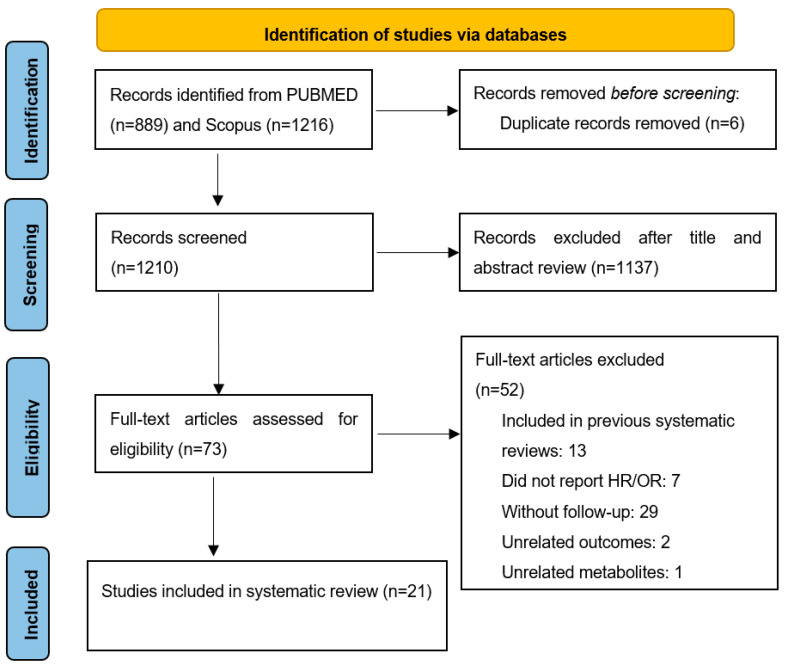
Flow diagram of studies assessed for eligibility per screening stage.

**Table 1 nutrients-14-02654-t001:** Characteristics of the 21 studies included in the systematic review.

First Author	Year/Country	Title	Journal	Study Design	N, Follow-Up Time	Baseline Characteristics of Participants	Assay Method and Metabolite Targets	Biological Sample	Main Outcome	Statistical Analysis	Covariates in Fully Adjusted Model	Adjusted HR or OR (95% CI)
Amrein, M. [25]	2022/Switzerland	Gut microbiota-dependent metabolite trimethylamine N-oxide (TMAO) and cardiovascular risk in patients with suspected functionally relevant coronary artery disease (fCAD)	Clin Res Cardiol	Cohort	1726 (223 deaths, 115 deaths due to CVD, 88 MI), 5 years	Patients with suspected functionally relevant CAD	LC-MS/MS; TMAO	Serum (non-fasting)	All-cause mortality, CVD mortality, MI	Cox regression	Age, sex, history of CAD, age, gender, BMI, smoking history, positive cardiovascular family history, hypertension, hypercholesterolemia, history of diabetes, history of stroke/TIA, history of CAD, previous MI, history of heart failure, cystatin-C	Per log2-transformed all-cause mortality HR 1.11 (95% CI 0.99–1.26), CVD mortality HR 1.19 (95% CI 1.01–1.40), MI HR 1.17 (95% CI 0.95–1.45)
Chen, Y.Y. [26]	2022/China	TMAO as a novel predictor of major adverse vascular events and recurrence in patients with large artery atherosclerotic ischemic stroke	Clin Appl Thromb Hemost	Cohort	291 (number of major vascular event cases not indicated), 3 months	Patients with ischemic stroke	LC-MS/MS; TMAO	Plasma (fasting); anticoagulant not indicated	Major vascular event (transient ischemic attack, recurrent ischemic stroke, MI)	Cox regression	Age, sex, hypertension, T2D, smoking, creatinine	>median vs. <median HR 3.12 (95% CI 1.02–9.61)
Fretts, A.M. [27]	2022/USA	Association of trimethylamine N-Oxide and metabolites with mortality in older adults	JAMA Network Open	Cohort	5333 (4791 total deaths), 13.2 years	20% of participants had CHD and 15% had T2D at baseline	LC-MS/MS; TMAO	Plasma (fasting); anticoagulant: EDTA	All-cause mortality	Cox regression	Age, sex, race, ethnicity, educational level, household income, smoking status, BMI, physical activity, treated hypertension, instrumental activities of daily living, self-reported health status, SBP, HDL-C, prevalent AF, prevalent CHD, history of MI, prevalent T2D, prevalent COPD, foods consumption	5th vs. 1st quintile all-cause mortality HR 1.30 (95% CI 1.17–1.44)
Liu, J. [23]	2021/USA	Gut microbiota–derived metabolites and risk of coronary artery disease: a prospective study among US men and women	Am J Clin Nutr	Nested case—control	1216 (608 CAD), not indicated	Participants were free of diabetes, cardiovascular disease, and cancer	LC-MS/MS; TMAO	Plasma (fasting and non-fasting); anticoagulant: EDTA	CAD	Conditional logistic regression	Age, sex, month of sample collection, fasting status at time of collection, smoking status, alcohol intake, physical activity, BMI, menopause status, family history of MI, aspirin use, T2D, hypertension, dyslipidemia, foods consumption	3rd vs. 1st tertile OR 1.23 (95% CI 0.89–1.70)
Lee, Y. [24]	2021/USA	Longitudinal plasma measures of trimethylamine N-Oxide and risk of atherosclerotic cardiovascular disease events in community-based older adults	J Am Heart Assoc	Cohort	4131 (1766 ASCVD), 15.0 years	Participants free of CVD at baseline	LC-MS/MS; TMAO	Plasma (fasting); anticoagulant: EDTA	ASCVD	Cox regression	Age, sex, race, study site, education, income, health status, smoking status, alcohol intake, physical activity, BMI, WC, lipid-lowering medication, antihypertensive medication, antibiotics, T2D, HDL-C, LDL-C, TG, CRP, SBP, DBP, diet, eGFR	5th vs. 1st quintile HR 0.98 (95% CI 0.80–1.20)
Frazier, R. [32]	2021/USA	Deoxycholic acid and risks of cardiovascular events, ESKD, and mortality in CKD: The CRIC study	Kidney Med	Cohort	3147 (512 CVD), 6.7 years; (575 HF), 7.0 years; (411 all-cause mortality), 7.9 years	Patients with chronic kidney disease but free of CVD at baseline	LC-MS/MS; deoxycholic acid	Serum (fasting)	CVD, HF, all-cause mortality	Cox regression	Age, sex, race, study site, eGFR, urinary protein, T2D, SBP, antihypertensive medications, smoking status, history of CVD, TChol, statin use, Il-6, CRP, fibroblast growth factor 23, parathyroid hormone, phosphate, calcium, albumin	>median vs. <median CVD HR 1.52 (0.74–3.12), HF 1.22 (95% CI 0.63–2.38), mortality HR 2.13 (95% CI 1.25–3.64)
Winther, S.A. [18]	2021/Denmark	Plasma trimethylamine N-oxide and its metabolic precursors and risk of mortality, cardiovascular and renal disease in individuals with type 2-diabetes and albuminuria	PLoS One	Cohort	311 (106 deaths 116 CVD, 44 deaths due to CVD), 21.9 years (6.8 years for death, 6.5 years for CVD)	Patients with T2D but free of CVD at baseline	LC-MS/MS; TMAO	Plasma (non-fasting); anticoagulant not indicated	CVD (cardiovascular mortality, non-fatal MI, ischaemic CVD, non-fatal stroke, amputation due to ischemia and cardiac or peripheral revascularization), CVD mortality and all-cause mortality	Cox regression	Age, sex, HbA1c, SBP, BMI, TChol, smoking, urinary albumin excretion rate and eGFR at baseline.	Per SD all-cause mortality HR 1.02 (95% CI 0.83–1.26), CV mortality 0.98 (95% CI 0.70–1.37), CVD 1.11 (95% CI 0.93–1.33)
Ringel, C. [19]	2021/Germany	Association of plasma trimethylamine N-oxide levels with atherosclerotic cardiovascular disease and factors of the metabolic syndrome	Atherosclerosis	3 Cohorts	1666 (99 deaths in LIFE-CAD), 9 years; (26 deaths in LIFE-AMI), 9 years; (194 deaths in CULPRIT-SHOCK), 30 days	Patients with ≥50% stenosis in at least one major coronary artery, and patients with angiographically normal coronary arteries (“LIFE-CAD”); patients with AMI (“LIFE-AMI”); patients with AMI with CS (“CULPRIT-SHOCK”)	LC-MS/MS; TMAO	Plasma (fasting status not indicated); anticoagulant: EDTA	All-cause mortality during 30 days and long term	Cox regression (LIFE-CAD, LIFE-AMI); logistic regression (CULPRIT-SHOCK)	Age, sex, BMI, T2D, eGFR, smoking status, SBP, DBP, high-sensitivity CRP, HDL-C, LDL-C, white blood cell count) and in LIFE-CAD additionally for presence of CAD.	Per SD (LIFE-CAD) HR 1.24 (95% CI 1.01–1.51), (LIFE-AMI) HR 1.07 (95% CI 0.70–1.63), (CULPRIT-SHOCK) OR 1.14 (95% CI 0.86–1.51)
Israr, M.Z. [20]	2021/UK	Association of gut-related metabolites with outcome in acute heart failure	Am Heart J	Cohort	806 (62 deaths), 30 days; (213 deaths), 1 year; (98 deaths/HF), 30 days; (313 deaths/HF), 1 year	Patients with acute HF at baseline	UHPLC-MS/MS; TMAO	Plasma (fasting status not indicated); anticoagulant: EDTA	All-cause mortality (death) and a composite of death and/or rehospitalization caused by HF (death/HF) at 30 days and 1 year	Cox regression	Sex, age, previous medical history (HF, ischemic heart disease, hypertension, and diabetes mellitus), NYHA class, smoking status, edema, AF, SBP, DBP, heart rate, hemoglobin, respiratory rate, blood sodium, and (log) N-terminal proBNP.	TMAO per SD death HR 1.39 (95% CI 1.05–1.84), 1 year HR 1.26 (95% CI 1.08–1.47); death/HF HR 1.38 (95% CI 1.10–1.73), 1 year HR 1.25 (95% CI 1.09–1.42)
Papandreou, C. [21]	2021/Spain	Choline metabolism and risk of atrial fibrillation and heart failure in the PREDIMED study	Clin Chem	2 Nested case-control	1127 (509 AF); 752 (326 HF), 10 years	Almost half of participants had T2D and were free of CVD at baseline	LC-MS/MS; TMAO	Plasma (fasting); anticoagulant: EDTA	AF and HF	Conditional logistic regression	Smoking, family history of premature CHD, physical activity, alcohol intake, BMI, intervention group, dyslipidemia, hypertension, T2D, medication use	TMAO per SD AF OR 1.04 (95% CI 0.92–1.17); HF OR 0.91 (95% CI 0.77–1.08) TMAO 4th vs. 1st quartile AF OR 1.02 (95% CI 0.72–1.44); HF OR 0.72 (95% CI 0.45–1.15)
Balasubramanian, R. [40]	2020/USA	Metabolomic profiles associated with all-cause mortality in the Women’s Health Initiative	Int J Epidemiol	2 Cohorts (discovery and replication sets)	943 (417 all-cause mortality (discovery)); 1355 (685 all-cause mortality (replication)), 10 years	10.9% and 14.8% of participants had T2D in the discovery and replication sets, respectively, and were free of CVD at baseline	LC-MS/MS; 470 metabolites (tryptophan)	Plasma (fasting); anticoagulant: EDTA	All-cause mortality	Cox regression	Age, WHI arm and CHD, BMI, SBP, hypertension treatment, T2D, smoking status, TChol, HDL-C	Tryptophan per SD WHI-HT HR 0.87 (95% CI 0.81–0.94); WHI-OS HR 0.82 (95% CI 0.75–0.89)
Croyal, M. [22]	2020/France	Plasma trimethylamine N-Oxide and risk of cardiovascular events in patients with type 2 diabetes	J Clin Endocrinol Metab	Cohort	1463 (403 MACEs and 538 deaths), 7.1 years	Patients with T2D at baseline	LC-MS/MS; TMAO	Plasma (non-fasting); anticoagulant not indicated	MACE (CV death, non-fatal MI and non-fatal stroke and all-cause mortality) and all-cause mortality	Cox regression	For MACE: sex, age, personal history of MI, eGFR, uACR, NT-proBNP; for all-cause mortality: sex, age, sinus rhythm, SBP, TNF receptor 1, angiopoietin-like 2	MACE 4th vs. 1st quartile HR 1.29 (95% CI 1.02–1.64), all-cause mortality 4th vs. 1st quartile HR 1.16 (95% CI 0.95–1.42)
Alonso, A. [29]	2019/USA	Serum metabolomics and incidence of atrial fibrillation (from the Atherosclerosis Risk in Communities Study)	Am J Cardiol	Cohort	3922 (608 AF), 20.4 years	14% of participants had T2D and were free of CVD at baseline	GC-MS/MS, LC-MS/MS; glycocholenate sulfate, glycolithocholate sulfate	Serum (fasting)	AF	Cox regression	Age, sex and race, smoking, BMI, SBP, use of antihypertensive medication, T2D, prevalent heart failure, and prevalent CHD, eGFR	Glycocholenate sulfate per SD HR 1.12 (95% CI 1.04–1.21) Glycolithocholate sulfate per SD HR 1.07 (95% CI 0.99–1.15)
Roncal, C. [38]	2019/Spain	Trimethylamine-N-oxide (TMAO) predicts cardiovascular mortality in peripheral artery disease	Sci Rep	Cohort	262 (135 MACE, 101 all-cause mortality and 39 cardiovascular mortality, 4 years	Patients with PAD at baseline	UHPLC-MS/MS; TMAO	Plasma (fasting); anticoagulant: citrate	All-cause mortality, CVD mortality, MACE (amputation, stroke, myocardial infarction and all-cause mortality)	Cox regression	Sex, age, hs-CRP, smoking, T2D, hypertension, dyslipidemia, HDL-C, eGFR (<60 mL/min/1.73 m2)	All-cause mortality per log unit HR 1.09 (95% CI 0.94–1.26), CV mortality TMAO per log unit HR 1.52 (95% CI 1.27–1.82), TMAO >2.26 μmol/L HR 3.36 (95% CI 1.68–6.70), MACE per log unit HR 1.08 (95% CI 0.95–1.23)
Fan, P.C. [31]	2019/Taiwan	Serum indoxyl sulfate predicts adverse cardiovascular events in patients with chronic kidney disease	J Formos Med Assoc	Cohort	147 (47 MACE), 3 years	Patients with chronic kidney disease but free of CVD at baseline	LC-MS/MS; indoxyl sulfate	Serum (fasting)	MACE (all-cause mortality, admission for HF and attack of acute coronary syndrome)	Cox regression	Age, sex, T2D, hypertension, SBP, left ventricular ejection fraction, hemoglobin, creatinine, albumin, hsCRP, LDL, TChol, medications	Per log unit HR 1.45 (95% CI 1.02–2.06)
Tobias, D.K. [39]	2018/USA	Circulating branched-chain amino acids and incident cardiovascular disease in a prospective cohort of US women	Circ Genom Precis Med	Cohort	27041 (2207 CVD), 18.6 years	Participants free of CVD at baseline	NMR; BCAAs	Plasma (fasting in 72.9%); anticoagulant: EDTA	CVD (MI, stroke, and coronary revascularization)	Cox regression	Age, randomized treatment assignments, fasting status at blood draw, menopausal status, current hormone therapy use, family history of MI, Caucasian race/ethnicity, smoking status, current 15+ c/d), AHEI diet quality score, alcohol intake, total physical activity MET-hrs/wk, cholesterol, hypertension, BMI	BCAAs per SD total CVD HR 1.13 (95% CI 1.08–1.18), MI HR 1.16 (95% CI 1.06–1.26); revascularization HR 1.17 (95% CI 1.11–1.25), stroke HR 1.07 (95% CI 0.99–1.15) Isoleucine per SD total CVD HR 1.14 (95% CI 1.09–1.19), MI HR 1.19 (95% CI 1.09–1.31); revascularization HR 1.24 (95% CI 1.16–1.32), stroke HR 1.05 (95% CI 0.97–1.13) Leucine per SD total CVD HR 1.06 (95% CI 1.02 -1.11), MI HR 1.05 (95% CI 0.97–1.15); revascularization HR 1.08 (95% CI 1.02–1.14), stroke HR 1.04 (95% CI 0.97–1.12) Valine per SD total CVD HR 1.13 (95% CI 1.08–1.18), MI HR 1.16 (95% CI 1.06–1.27 ); revascularization HR 1.16 (95% CI 1.09–1.24), stroke HR 1.06 (95% CI 0.99–1.15)
Huang, J. [30]	2018/USA	Serum metabolomic profiling of all-cause mortality: A prospective analysis in the alpha-tocopherol, beta-carotene cancer prevention (ATBC) study cohort	Am J Epidemiol	Cohort	620 (435 deaths (197 CVD and 107 cancer), 28 years	Participants free of CVD at baseline	UHPLC-MS/MS, GC-MS/MS; 406 metabolites (glycocholate, glycocholenate_sulfate, glycodeoxycholate, glycohyocholate, glycolithocholate_sulfate, glycoursodeoxycholate, taurocholenate_sulfate, taurodeoxycholate, taurolithocholate_3_sulfate, tauroursodeoxycholate )	Serum (fasting)	All-cause mortality	Cox regression	Age, BMI, number of cigarettes per day, TChol, HDL-C, hypertension, T2D, serum creatinine	No significant associations. HR with 95% CI were not indicated
Du, X. [41]	2018/China	Increased branched-chain amino acid levels are associated with long-term adverse cardiovascular events in patients with STEMI and acute heart failure	Life Sci	Cohort	138 (32 deaths and 21 hospitalizations for heart failure), 3 years	Patients with STEMI and AHF at baseline	LC-MS/MS; 26 amino acids (BCAAs)	Plasma (non-fasting); anticoagulant not indicated	Adverse cardiac events (composite of death and HF hospitalization)	Cox regression	Age, sex, history of T2D, history of hypertension, current smoking, and Killip class	Per SD HR 2.67 (95% CI: 2.41–4.41)
Yu, B. [36]	2016/USA	Associations between the serum metabolome and all-cause mortality among African Americans in the atherosclerosis risk in communities (ARIC) study	Am J Epidemiol	Cohort	1887 (671 deaths), 22.5 years	10.9% of participants had T2D and 6.3% had CVD at baseline	GC-MS/MS, LC-MS/MS; 600 metabolites (glycocholate)	Serum (fasting)	All-cause mortality, CVD mortality	Cox regression	Age, sex, BMI, SBP, antihypertensive medication use, diabetes status, current smoking status, prevalent cardiovascular disease status, HDL-C, TChol, and eGFR	Glycocholate per SD all-cause mortality HR 1.12 (95% CI 1.07–1.16), CV mortality HR 1.14 (95% CI 1.07–1.22)
Wang, C. [42]	2016/Taiwan	Increased p-cresyl sulfate level is independently associated with poor outcomes in patients with heart failure	Heart Vessels	Cohort	187 (35 composite event of death or HF-related re-hospitalization), 2.3 years	136 patients with HF and 51 participants without HF at baseline	LC-MS/MS; indoxyl sulfate	Plasma (non-fasting); anticoagulant: EDTA	Composite event of death or HF-related re-hospitalization	Cox regression	Age, LVEF, T2D, eGFR, and BNP	Indoxyl sulfate per μM HR 1.01 (95% CI 0.98–1.04)
Cheng, S. [28]	2015/USA	Distinct metabolomic signatures are associated with longevity in humans	Nat Commun	2 Cohorts (discovery and replication sets)	2327 (358 CVD, 439 all-cause mortality (discovery)), 13.6 years; 325 (36 all-cause mortality (replication)), 15.8 years	10% of participants had T2D and were free of CVD at baseline	LC-MS/MS; 217 metabolites (glycocholate, glycodeoxycholates, deoxycholates)	Plasma (fasting); anticoagulant: EDTA	CVD (coronary heart disease, HF, or stroke) and all-cause mortality	Cox regression	For CVD and all-cause mortality: age, sex, BMI, SBP, anti-hypertensive treatment, diabetes, smoking status, and total/HDL cholesterol	No significant associations. HR with 95% CI were not indicated

Abbreviations: AF, atrial fibrillation; AHEI, alternative healthy eating index; AMI, acute myocardial infarction; ASCVD, atherosclerotic cardiovascular disease; BCAAs, branched-chain amino acids (BCAAs); BMI, body mass index; CAD, coronary artery disease; CHD, coronary heart disease; CI, confidence interval; CRP, C-reactive protein; CVD, cardiovascular disease; DBP, diastolic blood pressure; eGFR, estimated glomerular filtration rate; EDTA, ethylenediaminetetraacetic acid; GC-MS/MS, gas chromatography–mass spectrometry; HbA1c, glycated hemoglobin; HDL-C, high-density lipoprotein cholesterol; HF, heart failure; HR, hazard ratio; hs-CRP, high-sensitivity C-reactive protein; HT, hormone trials; LC-MS/MS, liquid chromatography with tandem mass spectrometry; LDL, low-density lipoprotein; Log, logarithm; MACE, major adverse cardiac events; MET-hrs/wk, metabolic equivalent of task-hours per week; MI, myocardial infarction; NMR, nuclear magnetic resonance; NT-proBNP, N-terminal pro-brain natriuretic peptide; OR, odds ratio; OS, observational study; SBP, systolic blood pressure; SD, standard deviation; STEMI, ST-elevation myocardial infarction; TChol, total cholesterol; T2D, type 2 diabetes; TMAO, trimethylamine N-oxide; UHPLC-MS-MS, ultra-high performance liquid chromatography tandem mass spectrometry; WHI, Women’s health initiative.

## Data Availability

No new data were created or analyzed in this study. Data sharing is not applicable to this article.

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
