# Peer review of "Gut Microbiota-Derived Metabolites and Cardiovascular Disease Risk: A Systematic Review of Prospective Cohort Studies"

_nutrients, 2022, doi:10.3390/nu14132654_

Round 1

Reviewer 1 Report

The authors have conducted a systematic review of gut-microbiota derived metabolites and their association with CVD risk.

The approach to the review was systematic, well structured and well described and I think a review of this type will have general interest to the readers of this journal.

I would have liked to have seen more cross linking of results between studies being done as I read through the paper and not just the results of each paper being reiterated. In particular, I think there was more scope for the authors to discuss why studies may have conflicting results and differences in their sex, diet, ethnicity and other confounding factors. A discussion of whether confounding factors were assessed in each study would also have assisted interpretation. In general, naming each study using the lead author or the study name would have assisted the reader to follow the paper better.

At times, I also felt that certain chemical classes were being grouped together without consideration as to their origin or differences, or the title of the journal. For example, some of the bile acids discussed come from the liver, and are not formed as a result of microbial metabolism. Tryptophan was discussed, but is more likely to be of dietary origin than microbial.

Finally, it is not wrong to exclude other studies based on their previous inclusion in other systematic reviews, but a brief comparison of the findings of your review with the overall findings of previous reviews would have added more depth to the study.

Specific corrections.

Line 80: why were these specific metabolites selected for this study and others e.g. choline not. Why were metabolites which were not well covered in your search e.g. tryptophan ultimately included.

Line 165: since the coagulant used for plasma collection can influence results, please indicate this somewhere on the table or in the text.

Table 1: this table is a little crowded, and at least in the format I had, was crossing pages and not so easy to read. Possibly speak to the editorial team about making it a double page spread. Alternatively, perhaps spread column headers over more lines to allow more space in between headers. If room allows, I would encourage the male/female split and the plasma anti-coagulant to be included here.

Line 340: increasing evidence of a correlation does not improve the evidence of causation. Please re-word this section.

Author Response

Reply: We sincerely thank the Reviewer for appreciating our investigation, as well as for all the valuable comments and suggestions provided in the following lines, which have greatly improved the first version of the manuscript. We have addressed all of them in the following points, as well as in the manuscript, when required.

Please find below the itemized responses to the Reviewer’s comments.

I would have liked to have seen more cross linking of results between studies being done as I read through the paper and not just the results of each paper being reiterated. In particular, I think there was more scope for the authors to discuss why studies may have conflicting results and differences in their sex, diet, ethnicity and other confounding factors. A discussion of whether confounding factors were assessed in each study would also have assisted interpretation. In general, naming each study using the lead author or the study name would have assisted the reader to follow the paper better.

Reply: We tried to give a reasonable explanation of the inconsistent results among studies and following this suggestion we added a sentence in the limitations part about the differences in confounder adjustment across the studies, which may have resulted in different associations between metabolites (i.e. TMAO, bile acids) and CV outcomes. We also added a sentence in “4.1. TMAO” on the potential confounding effect of unmeasured factors such as dietary habits, gut microbiota composition and genetic variation on TMAO levels.

In the Results section we have included information about the lead author plus in some occasions the study name.

At times, I also felt that certain chemical classes were being grouped together without consideration as to their origin or differences, or the title of the journal. For example, some of the bile acids discussed come from the liver, and are not formed as a result of microbial metabolism. Tryptophan was discussed, but is more likely to be of dietary origin than microbial.

Reply: We would like to thank the Reviewer for this comment. Since primary bile acids are synthesized in the liver and then biotransformed by gut bacteria in the gastrointestinal tract to secondary bile acids, we excluded the results related to primary bile acids and kept those related to secondary bile acids in the revised manuscript and Table 1. We have also deleted the following paragraph from the Results section “3.4. Metabolites associated with CVD mortality and all-cause mortality”

Deleted paragraph: “Correspondingly, associations between taurocholate and all-cause mortality were reported in three other studies. Cheng et al. using two different cohorts, namely the Framingham Offspring Study (discovery set) and the Malmö Diet and Cancer study (replication set) quantified 217 plasma metabolites [25]. Over the total follow-up period of 13.6 years, there were 439 deaths from all causes in the discovery set and after 15.8 years there were 36 deaths in the replication set. In multivariable analyses and after Bonferroni correction, higher taurocholate levels were associated with all-cause mortality (HR 1.14 [95% CI 1.01-1.28]). However, these results were not replicated [25].”

We have also modified the paragraph named “4.3. Secondary bile acids” in the Discussion section.

We agree with the Reviewer that tryptophan is more likely to be of dietary origin as it is part of dietary proteins, and is thus high in protein-rich foods such as meat, fish, eggs, cheese, beans, and nuts. However, specific bacteria, such as Escherichia coli, are able to produce tryptophan. Therefore, we included this metabolite in the present systematic review.

Finally, it is not wrong to exclude other studies based on their previous inclusion in other systematic reviews, but a brief comparison of the findings of your review with the overall findings of previous reviews would have added more depth to the study.

Reply: Following this comment we have included in the Discussion section comparisons of our findings with those of previous systematic reviews. In the previous version of this manuscript we cited an earlier systematic review on the relationship between BCAAs and CVD.

Specific corrections.

Line 80: why were these specific metabolites selected for this study and others e.g. choline not. Why were metabolites which were not well covered in your search e.g. tryptophan ultimately included.

Reply: In the present systematic review we focus on common gut microbiota-derived metabolites like TMAO, secondary bile acids, SCFAs, BCAAs, tryptophan and indole derivatives that have been implicated in the pathogenesis of cardiovascular disorders and represent potential biomarkers for the early diagnosis and prognosis of these diseases. We did not select choline because it is not derived from gut microbial metabolism but it is metabolized by gut bacteria [PMID: 32764281]. We found one study on tryptophan and all-cause mortality and we believe that the inclusion of these results highlights the need for more work in this area.

Line 165: since the coagulant used for plasma collection can influence results, please indicate this somewhere on the table or in the text.

Reply: This is an important point. We have included this information in Table 1 under the column “Biological sample”.

Table 1: this table is a little crowded, and at least in the format I had, was crossing pages and not so easy to read. Possibly speak to the editorial team about making it a double page spread. Alternatively, perhaps spread column headers over more lines to allow more space in between headers. If room allows, I would encourage the male/female split and the plasma anti-coagulant to be included here.

Reply: We have tried to improve the readability of the Table 1 by reducing crossing and now each page starts with “Table 1. Cont”. As mentioned above we have included plasma anticoagulant in Table 1.

Line 340: increasing evidence of a correlation does not improve the evidence of causation. Please re-word this section.

Reply: We agree with this comment and we modified the sentence as: “These consistent findings reinforce the role of these metabolites as potential biomarkers of biological dysfunction related to CVD.”

Reviewer 2 Report

This is a systematic review manuscript. Authors discussed the roles of gut microbiota derived metabolites in cardiovascular disease risk. The data utilized was from 15 published prospective cohort studies. Authors concluded that the associations between BCAAs and CVD, indole derivatives and MACE and the inverse association between tryptophan and all-cause mortality look reproducible. On the prognostic value of TMAO in CVD risk, it does not seem reproducible, where authors gave reasonable explanation. The manuscript was well-drafted and organized. One concern need be addressed.

In page 4, authors mentioned they focused on 15 selected articles, which did not match the sentence in line #153-154, where totally 19 studies were mentioned. 

Author Response

We appreciate the positive comments of the Reviewer and his/her help to improve our manuscript.

In page 4, authors mentioned they focused on 15 selected articles, which did not match the sentence in line #153-154, where totally 19 studies were mentioned.

Reply: Thanks for this comment. We reduced the number of studies from 6 to 2 conducted in Taiwan.

Reviewer 3 Report

The present manuscript by Sanchez-Gimenez et al. provides a Systematic Review of Prospective Cohort Studies highlighting the role of gut-microbiota-derived metabolites on cardiovascular risks. Since the role of gut microbiota in cardiometabolic disorders is being established, this study is essential and timely. However, the manuscript has scope for substantial improvement.

1. Study design: Although the study is designed as a systematic review, it doesn't provide any quantitative/qualitative synthesis of the results. The authors need to meta-analyze the data sets and synthesize the comprehensive results.

            1.1.- Literature search needs to be done using multiple databases such as MEDLINE, EMBASE, SCOPUS, CINHL, and even google scholar. It will eliminate the chances of missing the critical literature. Furthermore, it is suggested to re-run the searches to include recent studies. 

             1.2. Articles listed in the previous studies [9,10,12,13,15–17,32] cant be excluded. However, new studies could be added to the existing ones to provide comprehensive outcomes.

2. Study lacks novelty as there are systematic reviews, metanalysis, and several narrative reviews in the literature (such as references 9,10,12,13,15–17,32). The best ways to improve the study are re-design (as per the previous comment), meta-analyzing the data under the set primary and secondary objectives, and showing results in the form of forest plots for each of the CVDs listed.

3. Discussion section can be written without subsections/headings.

4. Mechanisms behind the role of GMB-derived metabolites in CVDs should be discussed either in the introduction section or in the discussion section.

Author Response

The present manuscript by Sanchez-Gimenez et al. provides a Systematic Review of Prospective Cohort Studies highlighting the role of gut-microbiota-derived metabolites on cardiovascular risks. Since the role of gut microbiota in cardiometabolic disorders is being established, this study is essential and timely. However, the manuscript has scope for substantial improvement.

Reply: We sincerely thank the Reviewer for appreciating our investigation, as well as for all the valuable comments and suggestions provided in the following lines, which have greatly improved the first version of the manuscript. We have addressed all of them in the following points, as well as in the manuscript, when required.

Please find below the itemized responses to the Reviewer’s comments.

  1. Study design:Although the study is designed as a systematic review, it doesn't provide any quantitative/qualitative synthesis of the results. The authors need to meta-analyze the data sets and synthesize the comprehensive results.

Reply: This is an important point. Indeed, a meta-analysis of results across studies can produce useful findings. We were unable to conduct a meta-analysis, mainly because of the heterogeneity of the included articles, due to different metabolite targets, characteristics of the study populations and differential reporting of associations, increasing the possibility the results of the meta-analysis to be unreliable [PMID: 24320992]. We have acknowledged it as limitation and added a relevant sentence in the limitation part.

            1.1.- Literature search needs to be done using multiple databases such as MEDLINE, EMBASE, SCOPUS, CINHL, and even google scholar. It will eliminate the chances of missing the critical literature. Furthermore, it is suggested to re-run the searches to include recent studies. 

Reply: Our study was based on Medline interface (PubMed), which is considered the gold standard for biomedical database searching that includes more than 10 million references, abstracts of peer-reviewed journals [PMID: 11185333] and a wide range of databases, such as Cochrane.

As suggested by the Reviewer, we re-run the searches to include recent studies. Our updated literature search carried out up to 9 June 2022 yielded 889 non-duplicate records, 54 more than our previous search and finally 21 studies (6 new studies) were deemed eligible for inclusion in the present systematic review. Five of them examined TMAO, while 1 study analyzed the secondary bile acid deoxycholic acid. We have updated text, Tables and the flow diagram in the revised manuscript.

             1.2. Articles listed in the previous studies [9,10,12,13,15–17,32] can’t be excluded. However, new studies could be added to the existing ones to provide comprehensive outcomes.

Reply: Given that our systematic review is an attempt to update the available evidence on the associations of gut-microbiota derived metabolites with risk of several cardiovascular outcomes a topic that has been previously examined by several systematic reviews and since then many studies have been published we tried to update our search from these previous reviews. We have provided a brief comparison of the findings of our review with the overall findings of the previous reviews to give a more complete picture of this topic.

  1. Study lacks novelty as there are systematic reviews, metanalysis, and several narrative reviews in the literature (such as references 9,10,12,13,15–17,32). The best ways to improve the study are re-design (as per the previous comment), meta-analyzing the data under the set primary and secondary objectives, and showing results in the form of forest plots for each of the CVDs listed.

Reply: Our study is the first attempt to provides a comprehensive and updated systematic review about several gut-microbiota derived metabolites and risk of future CVD events. As we explained above we did not perform a meta-analysis.

  1. Discussion section can be written without subsections/headings.

Reply: Since many different metabolites and outcomes were analyzed in the present systematic review we have added subsections and headings in order to make easier to follow.

  1. Mechanisms behind the role of GMB-derived metabolites in CVDs should be discussed either in the introduction section or in the discussion section.

Reply: We have added several potential mechanisms explaining the relationships between the microbial metabolites and cardiovascular events in the Discussion section.

Round 2

Reviewer 3 Report

The Authors have improved the manuscript; however, some concerns remained unanswered.

1. Reply to previous comment 1.1 is unsatisfactory. Medline is one of the best databases to perform a literature search, but other databases are recommended to avoid missing any important information (article).

2. Reply to comment #2 is also not satisfactory.

3. Please update the study numbers in table 1 i.e., 21.

Author Response

The Authors have improved the manuscript; however, some concerns remained unanswered.

We appreciate the positive comments of the Reviewer and his/her help to improve our manuscript.

  1. Reply to previous comment 1.1 is unsatisfactory. Medline is one of the best databases to perform a literature search, but other databases are recommended to avoid missing any important information (article).

Reply: Following this comment we run literature search using the SCOPUS and identified 1,216 records. After removing 6 duplicates, we screened 1,210 records but we did not identify studies that were not included in PubMed. We have modified the manuscript and supplementary file accordingly.

  1. Reply to comment #2 is also not satisfactory.

Reply: Our updated and comprehensive systematic review of gut-microbiota derived metabolites in relation to CV outcomes identified 11 new studies on the relationship between TMAO and CV outcomes, 2 new studies on the associations between BCAAs and CV outcomes and for the first time, synthesized studies related to bile acids, indole derivatives and CV events. Therefore, the novelty of this systematic review lies in the combination of updating the information on already examined gut-microbiota derived metabolites and other gut-microbiota derived metabolites that have never been qualitatively synthesized before.

Indeed, a meta-analysis of results across studies can produce useful findings. We were unable to conduct a meta-analysis, mainly because of the heterogeneity of the included articles, due to different metabolite targets, characteristics of the study populations and differential reporting of associations, increasing the possibility the results of the meta-analysis to be unreliable [PMID: 24320992]. We have acknowledged it as limitation and added a relevant sentence in the limitation part.

  1. Please update the study numbers in table 1 i.e., 21.

Reply: We present the study numbers in Table 1 following the chronological order of their publication.